# Recalibrating probabilistic forecasts of epidemics

**Aaron Rumack**[1]*, **Ryan J. Tibshirani**[1,2], **Roni Rosenfeld**[1]

**1** Machine Learning Department, Carnegie Mellon University, Pittsburgh, Pennsylvania, United States of America, **2** Department of Statistics & Data Science, Carnegie Mellon University, Pittsburgh, Pennsylvania, United States of America

* arumack@andrew.cmu.edu

## Abstract

Distributional forecasts are important for a wide variety of applications, including forecasting epidemics. Often, forecasts are miscalibrated, or unreliable in assigning uncertainty to future events. We present a recalibration method that can be applied to a black-box forecaster given retrospective forecasts and observations, as well as an extension to make this method more effective in recalibrating epidemic forecasts. This method is guaranteed to improve calibration and log score performance when trained and measured in-sample. We also prove that the increase in expected log score of a recalibrated forecaster is equal to the entropy of the PIT distribution. We apply this recalibration method to the 27 influenza forecasters in the FluSight Network and show that recalibration reliably improves forecast accuracy and calibration. This method, available on Github, is effective, robust, and easy to use as a post-processing tool to improve epidemic forecasts.

**Data Availability Statement:** The code and data are available at https://github.com/rumackaaron/recalibration.

**Funding:** AR was supported by a fellowship from the Center for Machine Learning and Health at Carnegie Mellon University (https://www.cs.cmu.

## Author summary

Epidemics of infectious disease cause millions of deaths worldwide each year, and reliable epidemic forecasts can allow public health officials to respond to mitigate the effects of epidemics. However, because epidemic forecasting is a difficult task, many epidemic forecasts are not calibrated. Calibration is a desired property of any forecast, and we provide a post-processing method that recalibrates forecasts. We demonstrate the effectiveness of this method in improving accuracy and calibration on a wide variety of influenza forecasters. We also show a quantitative relationship between calibration and a forecaster's expected score. Our recalibration method is a tool that any forecaster can use, regardless of model choice, to improve forecast accuracy and reliability. This work provides a bridge between forecasting theory, which rarely deals with applications in domains that are new or have little data, and some recent applications of epidemic forecasting, where forecast calibration is rarely analyzed systematically.

edu/cmlh-cfp). RR and AR were supported by
McCune Foundation grant FP00004784 (https://
www.mccune.org). RT and RR were supported by
Centers for Disease Control and Prevention grant
U01IP001121 (https://www.cdc.gov). The funders
had no role in study design, data collection and
analysis, decision to publish, or preparation of the
manuscript.

**Competing interests:** The authors have declared
that no competing interests exist.

## 1 Introduction

Epidemic forecasting is an important tool to inform the public health response to outbreaks of infectious diseases. Often, decision makers can take more effective action with an estimate of the uncertainty in a forecasted target. For this reason, distributional forecasts are more desirable than point forecasts. A distributional forecast is a probability distribution over the target variable and measures the uncertainty in the prediction, as opposed to a point forecast, which is just a scalar value for each target and has no measure of uncertainty. A desired property of distributional forecasts is *calibration*, or reliability between forecasts and the true distribution of the variable forecasted (a mathematical definition is given in Section 2). Along with uncertainty and resolution, calibration is one of three components of a forecaster's accuracy as measured by any proper score [1], with better calibration resulting in a better score. It is therefore important for a forecaster to produce calibrated forecasts.

Previous work has described general forecasting theory and calibration and evaluated the calibration of certain forecasts [2–5]. Later work has gone beyond just describing calibration, presenting post-processing algorithms to recalibrate forecasts that were previously miscalibrated. Nonparametric techniques for recalibration of ensemble forecasts include rank histogram correction [6], Bayesian model averaging [7], linear pooling [8], and probability anomaly correction [9]. Brocklehurst et al. [10] provide a nonparametric approach using the empirical CDF, which can recalibrate any forecast of a scalar target. Parametric approaches include logistic regression [11], extended linear regression [12] and beta-transform linear pooling [8]. Wilks and Hamill [13] compare the performance of different recalibration techniques for different meteorological targets with different amounts of training data.

Much of the work in recalibration has been applied to weather forecasting, and thus many of the techniques are not applicable in other forecasting domains. The most popular weather forecasting models create a distribution from a series of point predictions, with each point being the result of a simulation under varying initial conditions. Many of the existing recalibration methods are defined only for this type of ensemble forecaster. For example, Bayesian model averaging assumes that an ensemble forecast is comprised of the same *N* forecasts in each observation. This method cannot be extended trivially to a domain where the forecaster itself outputs a distribution. Additionally, weather forecasts usually have a plethora of training data on which to train recalibration methods. For example, recalibration has been applied to a set of weather forecasts generated daily from 1979 to at least 2006, almost 10,000 days [14]. In settings like these, techniques need not be robust to small amounts of recalibration training data.

To be clear on nomenclature, throughout this paper, we use the term *forecast* to refer to the predicted probability distribution of a variable and the term *forecaster* to refer to an algorithm that produces a forecast for a variable given a context. Common examples of forecasters are an algorithm that forecasts the amount of precipitation two days in advance given current meteorological information, one that forecasts the price of a certain stock given the stock's historical trend, or one that forecasts the statewide influenza incidence given historical incidence data. We also distinguish between *calibration* and *recalibration*; calibration refers to the property of a forecaster, and recalibration refers to a method whose goal is to make a forecaster more calibrated. Specifically, recalibration takes as input a set of a forecaster's forecasts and corresponding observations ("training data"), and outputs a forecaster which should be more calibrated on a different set of forecasts and observations ("test data").

In what follows, we present a generalized approach to forecast recalibration and show its performance when applied to forecasters in the FluSight Network. We demonstrate that across the diverse set of FluSight forecasters, recalibration consistently improves not just calibration but accuracy as well.

## 2 Methods

Consider the following setup. At each $i = 1, 2, 3\ldots$, a forecaster $M$ outputs a density forecast $f_i$ given features $x_i$ for a continuously distributed scalar random variable $y_i$ whose true distribution is $h_i$. As a regularity condition, we assume that the corresponding cumulative distribution functions (CDFs) $F_i$ and $H_i$ are continuous and strictly increasing. The forecaster $M$ is evaluated according to a proper scoring rule, such as the quadratic score [15] or the logarithmic score [16].

The goal of a forecaster is to produce ideal forecasts, i.e., to forecast $f_i = h_i$, the true distribution of $y_i$, for each $i$, though this is usually unattainable. We can inspect how close a forecaster is to being ideal with the distribution of the probability integral transform (PIT) values [17]. For each forecast $f_i$ and observed value $y_i$, the PIT is defined as

$$\text{PIT}(f_i, y_i) = F_i(y_i),$$

where $F_i$ is the CDF of $f_i$. A necessary (but not sufficient) condition for a forecaster to be ideal is *probabilistic calibration* [3]:

$$\frac{1}{N} \sum_{i=1}^{N} H_i \circ F_i^{-1}(p) \to p \ \text{ as } N \to \infty, \ \text{ for all } p \in (0, 1).$$

(Here and throughout we interpret convergence in the almost sure sense.) An example of a probabilistically calibrated forecaster that is not ideal is the so-called climatological forecaster, which for each $i$ outputs the marginal distribution of $y_i$ over $i = 1, 2, 3, \ldots$. To make this concrete, suppose each $y_i$ is distributed as $\mathcal{N}(\mu_i, 1)$, a normal distribution with mean $\mu_i$ and variance 1, and each $\mu_i$ itself follows $\mathcal{N}(0, 1)$, then the climatological forecaster simply outputs $\mathcal{N}(0, 2)$ for each $i$.

Note that the PIT distribution of a probabilistically calibrated forecaster is close to uniform in large samples. The expected CDF of the PIT distribution is

$$G(p) = \mathbb{E}[\mathbb{P}[F_i(y_i) \leq p]] = \mathbb{E}[\mathbb{P}[y_i \leq F_i^{-1}(p)]] = \mathbb{E}[H_i \circ F_i^{-1}(p)],$$

where here $\mathbb{E}$ denotes the sample average operator over $i = 1, \ldots, N$. This expression converges to $p$ as $N \to \infty$ when the forecaster is probabilistically calibrated. Thus an examination of the distribution of PIT values—looking for potential deviations from uniformity—serves as a good diagnostic tool to assess probabilistic calibration. Many use a PIT histogram to examine the PIT distribution because it is easy to read and understand [3]. For example, if the PIT distribution is bell-shaped, then the forecaster does not put enough weight in the middle of its distribution and is underconfident. In general, we can compare the PIT density to the horizontal line at 1, which corresponds to the uniform density. The greater the deviation from this line (which can be quantified via Kullback-Leibler divergence from the uniform distribution to the PIT distribution, or equivalently, negative entropy of the PIT distribution), the greater the miscalibration; see Fig 1 for examples.

Our recalibration method uses $G$ as a CDF-CDF transform. The recalibrated forecaster, denoted $M^*$, is defined by a recalibrated forecast CDF of $F_i^*(y) = G(F_i(y))$, for each $i$. By the chain rule, the recalibrated forecast density is $f_i^*(y) = g(F_i(y)) \cdot f_i(y)$, for each $i$. Thus the recalibrated forecast $f_i^*$ is the original forecast $f_i$ weighted by the PIT density $g$. An illustration of this method is provided in Fig 2. In practice, of course, we do not have access to the true distributions $H_i$, so we need to estimate $G$ from PIT values. A key assumption is that the PIT distribution of the training forecasts is the same as that of the test forecasts. Otherwise, applying $G$ as a CDF-CDF transform will not produce probabilistically calibrated forecasts. The ultimate

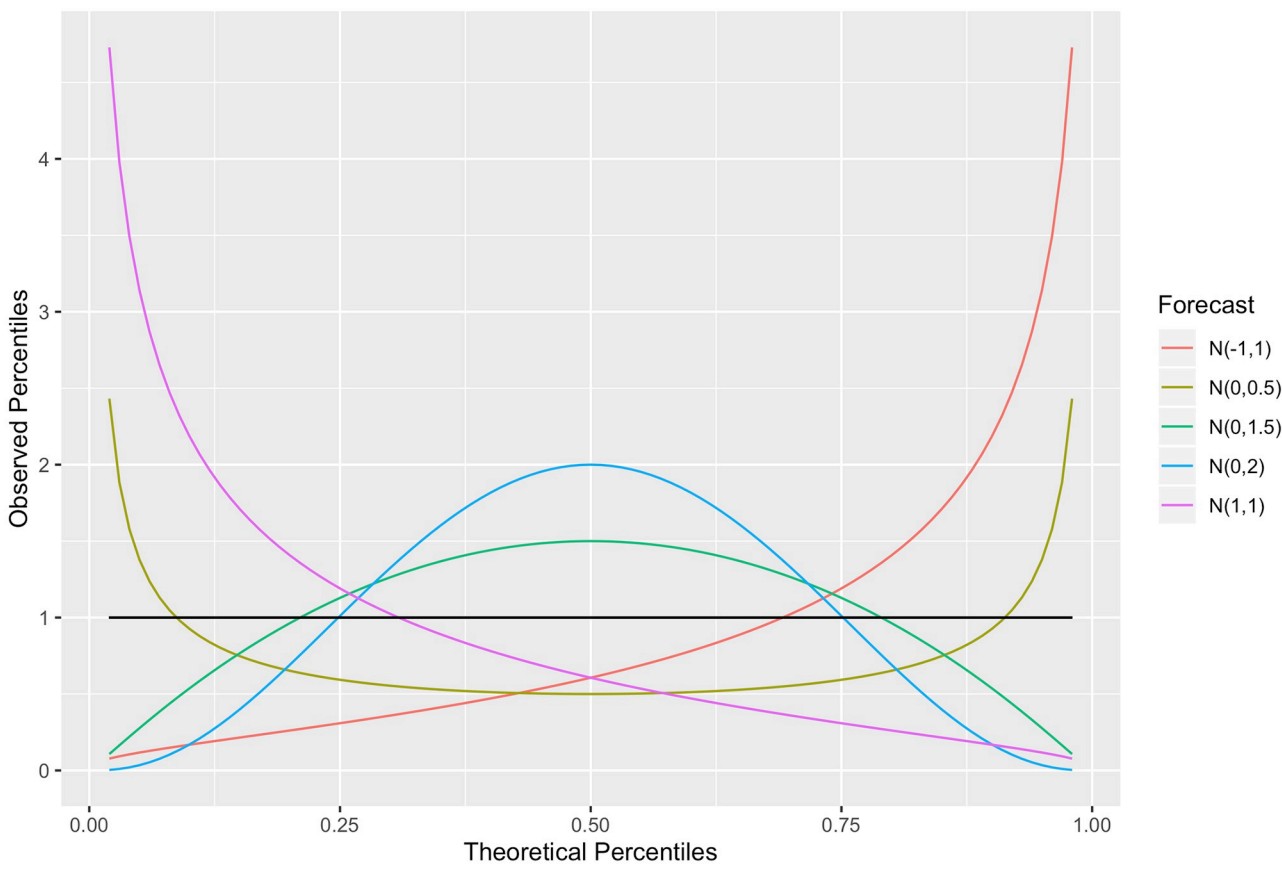

**Fig 1. Densities of PIT distributions for five sample forecasters, when the true distribution is a standard normal.**

estimate of $G$ that we propose in this paper will be an ensemble (weighted linear combination) of three estimates: a nonparametric method, a parametric method, and a null method. First, we will motivate calibration as a tool to increase forecast accuracy, and then, we explain the individual estimation methods.

### 2.1 Calibration and log score

In order to quantify how well a forecaster is calibrated, we calculate the entropy of the distribution of PIT values. As above, $G$ is the CDF of the PIT distribution of $M$. The entropy of the PIT density $g$ is defined as

$$H(g) = -\int_{p=0}^{1} g(p) \log g(p)\, dp.$$

If $M$ is probabilistically calibrated, then (asymptotically, as $N \to \infty$) the PIT values are uniform and the entropy is zero because $g(p)$ is 1 everywhere. When the PIT values are not uniform, the entropy is negative.

Entropy is also useful because it provides an understanding of how miscalibration penalizes the expected log score, as shown below. First observe that

$$g(p) = \frac{d}{dp} G(p) = \frac{d}{dp} \mathbb{E}[H_i \circ F_i^{-1}(p)] = \mathbb{E}\left[\frac{h_i(F_i^{-1}(p))}{f_i(F_i^{-1}(p))}\right],$$

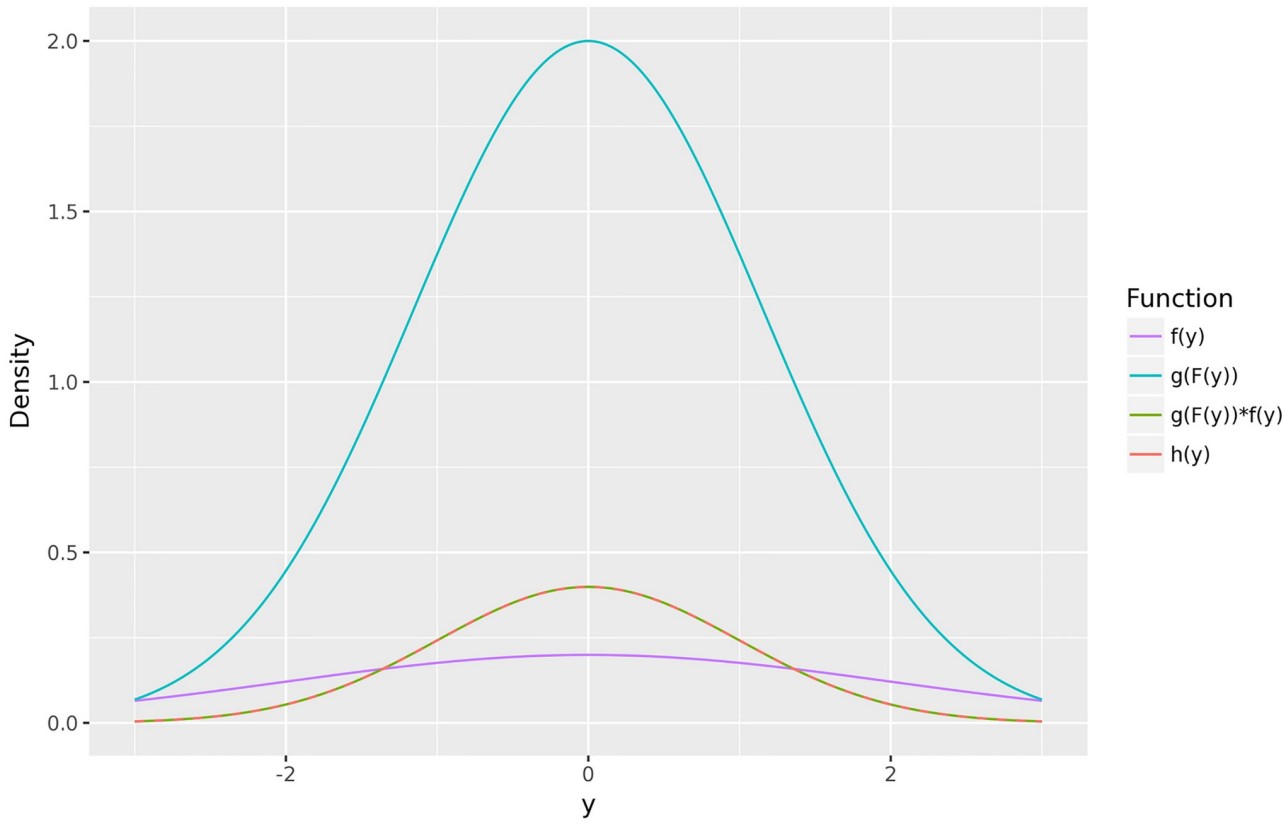

**Fig 2. An illustration of recalibration.** The original, underconfident forecast density is $f(y) = \mathcal{N}(0, 2)$ while the true density is $h(y) = \mathcal{N}(0, 1)$. By calculating the PIT density $g$ and producing a recalibrated forecast as the product $g(F(y)) \cdot f(y)$, we recover the true $h(y)$.

where the last step assumes the smoothness and integrability conditions on $h_i, f_i$ needed to exchange expectation and differentiation (the Leibniz rule). Next observe that

$$
\begin{aligned}
\mathbb{E}[\log f_i^*(y_i)] - \mathbb{E}[\log f_i(y_i)] &= \mathbb{E}[\log g(F_i(y_i))] \\
&= \mathbb{E}\left[\int_{-\infty}^{\infty} \log g(F_i(y_i)) h_i(y_i)\, dy_i\right] \\
&= \mathbb{E}\left[\int_0^1 \log g(F_i(F_i^{-1}(p))) \frac{h_i(F_i^{-1}(p))}{f_i(F_i^{-1}(p))}\, dp\right] \\
&= \int_0^1 \mathbb{E}\left[\log g(p) \frac{h_i(F_i^{-1}(p))}{f_i(F_i^{-1}(p))}\right] dp \\
&= \int_{p=0}^1 g(p) \log g(p) = -H(g),
\end{aligned}
\tag{1}
$$

where the third line is obtained by a variable substitution, and fourth by applying the Leibniz rule again assuming the needed regularity conditions.

For any forecaster, if the PIT distribution is the same for the training data and the test data, then the improvement of the recalibrated forecast's log score can be estimated by estimating the negative entropy of $g$ (note that the entropy of any distribution on $[0, 1]$ is nonpositive).

We can explain this intuitively as well: the more negative $H(g)$ is, the more it indicates that there is information lying in the structure of $g$ that can be extracted to improve forecasts.

## 2.2 Nonparametric correction

Given an observed training set of PIT values for a forecaster, $F_i(y_i)$, $i = 1, \ldots, N$, the empirical PIT CDF is

$$\hat{G}(x) = \frac{1}{N} \sum_{i=1}^{N} \mathbb{I}[F_i(y_i) \leq x].$$

As $\hat{G}$ is discrete, it does not admit a well-defined density, and hence to use this for recalibration we can first smooth $\hat{G}$ using a monotone cubic spline interpolant, and then it will have a bonafide density $\hat{g}$, which is itself smooth (twice continuously differentiable, to be precise). Using this for recalibration produces $f_i^*(y) = \hat{g}_i(F_i(y)) \cdot f_i(y)$.

In practice, with a large amount of training data, recalibration using the empirical CDF as described above can be effective. However, with little training data, or a lot of diversity within the training data among the distributions of $y_i$, it can be ineffective for assuring calibration on the test set. This is in line with the practical difficulties of using nonparametric, distribution-free methods in general.

## 2.3 Parametric correction

Gneiting and Ranjan [8] present a recalibration method originally motivated by redistributing weights on the components of an ensemble forecast, but their method can applied generally to recalibrate any black box forecaster. Given an observed training set of PIT values, $F_i(y_i)$, $i = 1, \ldots, N$, we fit a beta density $\hat{g}$ via maximum likelihood estimation. This in fact corresponds to the beta transform that maximizes the log score of the recalibrated forecaster on the training data [8].

This parametric model is more resilient to minimal training data, and a beta distribution is usually an effective estimate of the PIT distribution: because a beta density can be either convex or concave, it is flexible enough to fit the PIT distribution of overconfident and under-confident forecasters; and because the mean can be in the interval (0, 1), it can fit biased fore-casters as well. However, problematic behaviors arise at the tails. Except in exceptional cases (one or both of its two shape parameters is exactly 1), the beta density is 0 or $\infty$ at the end-points of its support, which can cause problems for recalibration (there can be a big gap between the true PIT density and $\hat{g}$ in the tails).

## 2.4 Null correction

The final component of the recalibration ensemble is a null correction, in which there is no recalibration at all, i.e., we simply set $f_i^*(y) = f_i(y)$. This prevents overfitting and decreases variance of the overall ensemble correction, to be described next.

## 2.5 Recalibration ensemble

The final recalibration system uses the three components described previously and weights them in an ensemble. The ensemble weights are calculated to maximize the overall log score. Letting $f_{ij}^*$ denote the forecast density for sample $i$ and component $j$, the weights ensemble $w$

are defined by solving the optimization problem:

$$\underset{w}{\text{minimize}} \quad \frac{1}{N} \sum_{i=1}^{N} \log \left( \sum_{j=1}^{P} w_j f_{ij}^*(y_i) \right) \quad \text{subject to} \quad w \geq 0, \ \sum_{j=1}^{P} w_j = 1, \tag{2}$$

where $p$ is the number of ensemble components (for us, $p = 3$) and the constraint $w \geq 0$ is to be interpreted componentwise.

A component's weight in the ensemble is not necessarily proportional to that component's performance. For example, if the two best components are very similar to each other, one may have a very small weight because that component's information is effectively represented by the other component.

## 2.6 Recalibration under seasonality

Epidemic forecasting presents a new challenge for recalibration. The methodology discussed above assumes that the previous behavior of a forecaster is indicative of future behavior, or more concretely, that the PIT distribution on the training set will be similar to that on the test set. However, this is not necessarily the case in epidemic forecasting, due to the fact that a forecaster's behavior generally changes across the different phases of an epidemic. For example, some forecasters do not predict enough of a change in disease incidence from one week to the next. For such a forecaster, the PIT values are usually too high between a season's onset and peak, because incidence increases more quickly than forecasted. Conversely, after the season peaks, the PIT values are too low, because incidence decreases more quickly than forecasted.

In order to account for such nonstationarity in the PIT distribution, we would like to form and use a special training set based on forecasts made at similar points in the epidemic curve in different seasons. This is not a straightforward task to do in real-time, since one cannot always be sure whether the peak has passed yet or not. However, for seasonal epidemics, we can take advantage of seasonality and build this training set based on the calendar weeks in which the forecasts were made. For example, a forecast made in week 6 can be recalibrated based on forecasts in other seasons made in weeks in between 3 and 9. This is what we do in our experiments in this paper, with more details given in the next section.

## 3 Results

We apply this ensemble recalibration method to data from influenza forecasting in the US. In an effort to better prepare for seasonal influenza, the US CDC has organized a seasonal influenza forecasting challenge every year since 2013, called the FluSight Challenge [18]. In 2017, a group of forecasters formed the FluSight Network [19] and began submitting an ensemble forecast of 27 component forecasters. As part of this collaboration, each of these forecasters produced and stored retrospective forecasts spanning 9 seasons, from 2010–11 to 2018–19. The retrospective forecasts were produced at the same time, with each forecaster using the same method for all seasons. Had the forecaster modified its algorithm from season to season, the previous forecast performance would not be predictive of future forecast performance, violating the assumptions behind this recalibration method. These forecasters include mechanistic and non-mechanistic forecasters, as well as baseline forecasters. They are diverse in behavior, accuracy, and calibration, and therefore provide an interesting challenge for our recalibration method, which treats the forecaster as a black box.

First, we summarize the retrospective forecasts in the FluSight data set. Each week, a forecast is produced for seven forecasting targets, all of which are based on weighted ILI (wILI), a population-weighted average of the percentage of outpatient visits with influenza-like illness derived from reports to the CDC from a network of healthcare providers called ILINet [20]. The forecasting targets are:

- season onset (the first week where wILI is above a predefined baseline for three consecutive weeks);

- season peak week (week of maximum wILI);

- season peak percentage (maximum wILI value);

- the wILI value at 1, 2, 3, and 4 weeks ahead of the current week.

The first three targets are referred to as seasonal targets and the last four targets are referred to as short-term targets. Each forecast is discretized over predetermined bins, forming a histogram distribution. For the season onset and season peak week targets, the width of each bin is one week, and for the other targets, the width of each bin is 0.1% wILI. Forecasts are produced for each of the 10 HHS Regions as well as the US as a whole, for a total of 9 seasons, from 2010–11 to 2018–19. Thus to be clear, the forecasts in this FluSight data set are indexed by forecaster, target, season, forecast week, and location.

Next, we describe the training setup we use for recalibrating the forecasts in this data set, which is a kind of nested leave-one-season-out cross-validation. This is laid out in the steps below for a given forecaster and forecasting target, and a particular season $s$.

1. Create recalibrated forecasts for all seasons $r \neq s$, using each of the three methods: nonparametric, parametric, and null. For a forecast in season $r$ at week $i$ and at location $\ell$, we build a training set using PIT values from all seasons other than $r$ and $s$, all available forecast weeks in $[i - 3, i + 3]$ (within three weeks of $i$), and all locations. These recalibrated forecasts are only used for training the ensemble weights in the following step.

2. Optimize the ensemble weights $w$ by solving (2) using the recalibrated forecasts from Step 1.

3. Create recalibrated forecasts for season $s$, again using each of the three methods: nonparametric, parametric, and null. This is just as in Step 1, except we use one more season in the training set. Explicitly, for a forecast in season $s$ at week $i$ and at location $\ell$, we build a training set using PIT values from all seasons other than $s$, all forecast weeks in $[i - 3, i + 3]$ (within three weeks of $i$), and all locations.

4. Create ensemble recalibrated forecasts from season $i$, using the recalibration components from Step 3 and the weights from Step 2.

In what follows, we present and discuss the results. The code and data used to produce all of these results is publicly available online [21].

### 3.1 Effect of varying window size

The training procedure just presented assumes a window of $k = 3$ weeks on either side of a given week $i$ in order to build the set of PIT values used for recalibration (using forecast data from other seasons). However, we could consider varying $k$, which would navigate something like a bias-variance tradeoff. We would expect the optimal window $k$ to be larger for the nonparametric recalibration method versus the parametric one. It turns out that $k = 3$ is typically a reasonable choice for both, as displayed in Fig 3.

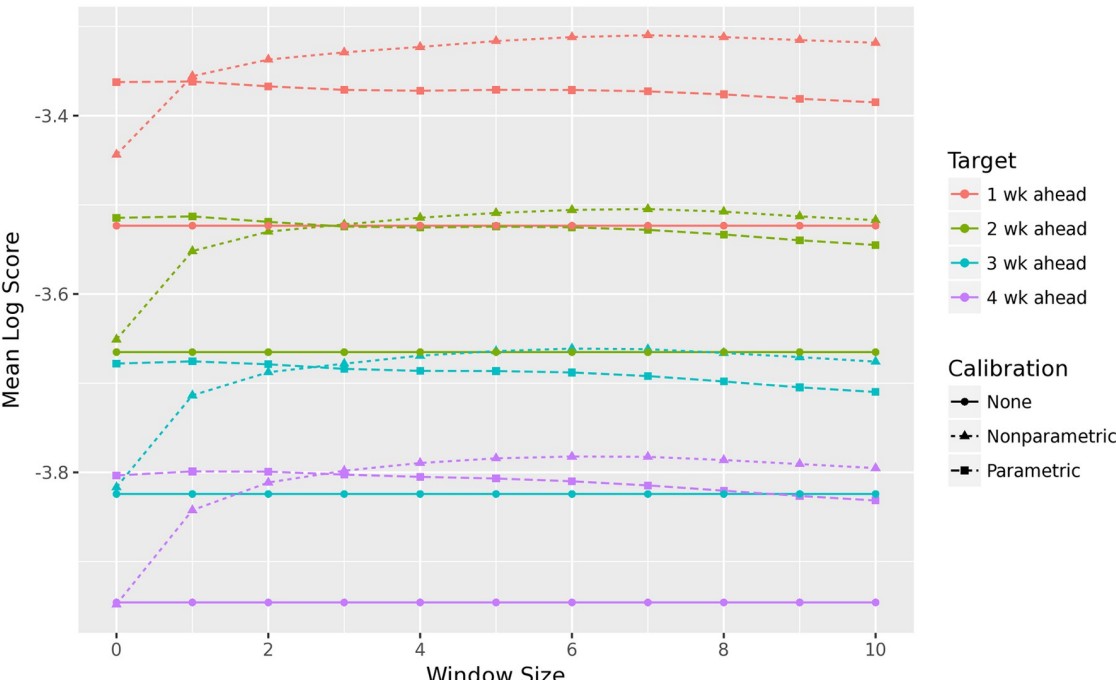

**Fig 3. Mean log score, averaged over all forecasters, for the different recalibration methods.** A window size of *k* corresponds to training recalibration on forecasts within *k* weeks of the given forecast week, where available, inclusive. Log score is averaged over 9 seasons, 11 locations, and 29 weeks (higher log score is better). The largest window sizes slightly hurt the performance of the parametric model, and the smallest window sizes significantly hurt the nonparametric model. Averaged over all forecasters, the improvement in performance due to calibration is roughly equal to the improvement in performance by reducing the forecast horizon by a week.

### 3.2 Forecast accuracy and calibration

For the short-term targets, the ensemble recalibration method improves the mean log score for almost all forecasters. Both the nonparametric and parametric recalibration methods significantly improve the mean log score, and the ensemble improves it even further. For the seasonal targets, some component recalibration methods do not improve accuracy, although the ensemble method does improve accuracy, averaged over all forecasters. However, the ensemble improves accuracy for seasonal targets in only about three-quarters of forecasters. See Figs 4 and 5.

Fig 6 gives a more direct comparison of improvements in accuracy versus calibration, i.e., in mean log score versus entropy, for the short-term forecasts. (Note that we estimate the entropy of the distribution of PIT values using a simple histogram estimator with 100 equal bins along the interval [0, 1].) We see a clear linear trend, with slope approximately 1, confirming our expectations from (1).

Finally, in Fig 7, we show that our ensemble recalibration method increases the entropy of the PIT distribution to nearly zero for nearly every forecaster. The two exceptions, the line segments towards the bottom of Fig 7, correspond to particularly poor forecasters (so poor that are outperformed by a baseline forecaster that outputs a uniform distribution).

### 3.3 Effect of number of training seasons

We chose to apply our recalibration to the FluSight Challenge because there are many forecasters available over many seasons for testing and training. When recalibrating forecasts of other

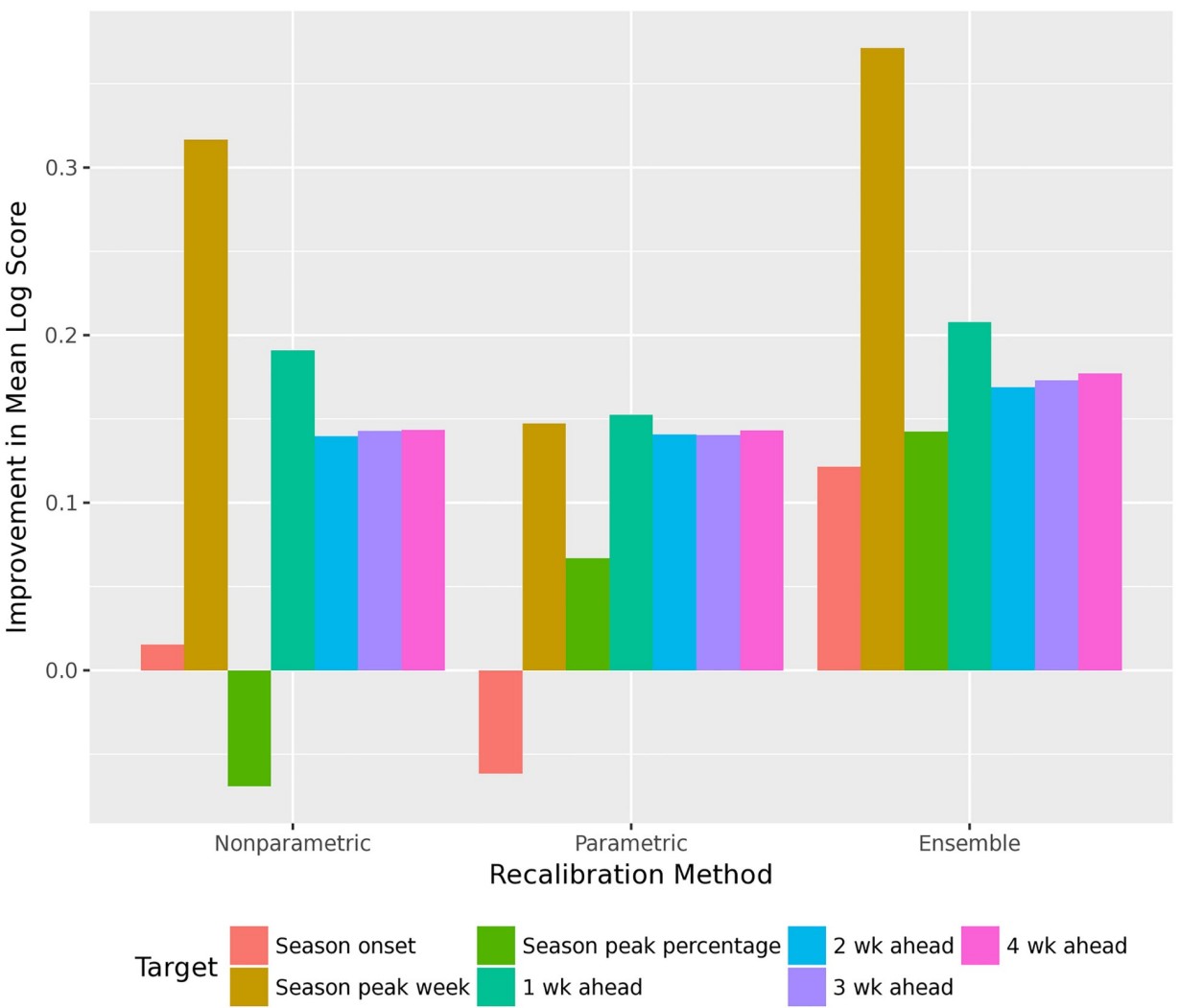

**Fig 4. Improvement in mean log score, for the different recalibration methods.** Log score is averaged over all 27 forecasters in the FluSight, 9 seasons, 11 locations, and 29 weeks (higher log score is better). The ensemble recalibration method improves accuracy for every target.

epidemics, there may be significantly less training data available. Fortunately, these methods are robust to recalibrating FluSight Challenge forecastssituations with little training data. The parametric recalibration method improves the mean log score, averaged over all 27 forecasts, with just two training seasons, and the nonparametric recalibration improves average performance with four training seasons, as shown in Fig 8.

Because we train selectively based on seasonality, as discussed in Section 2.6, each training season and location contributes only 7 PIT values to estimate $G$. We pool 11 locations together, so the parametric method can improve performance with roughly 150 PIT values, and the nonparametric method can improve performance with roughly 300 PIT values.

### 3.4 Recalibrating the FluSight ensemble

As we just saw, recalibration improves the performance of the individual forecasters in the Flu-Sight Network. A natural follow up is therefore to investigate whether it can improve the

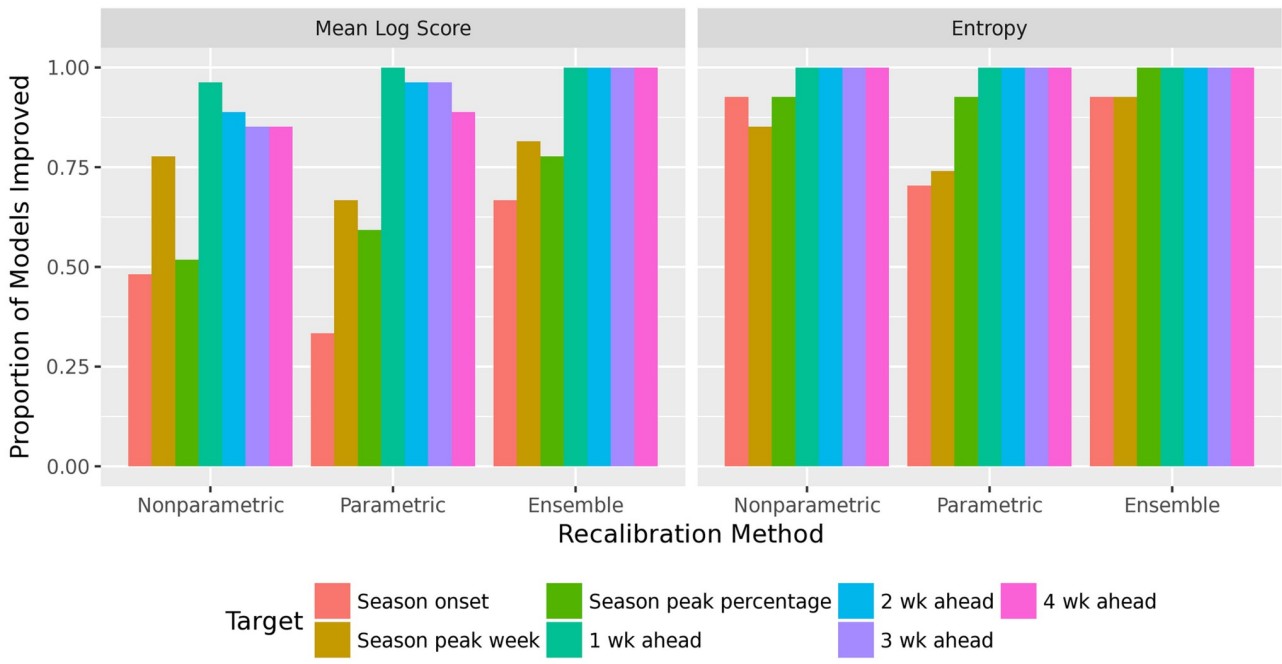

**Fig 5. Proportion of forecasters for which recalibration improves mean log score (left) and entropy of the PIT values (right).** The ensemble method improves accuracy for the short-term targets for all forecasters, and most forecasters for the seasonal targets. It also improves calibration (as measured by entropy) for most forecasters and most targets. The ensemble method outperforms both the nonparametric and parametric methods.

performance of the FluSight ensemble, a forecaster that combines 27 component forecasters (the individual FluSight forecasters), whose construction is described in [19].

As both recalibration and ensembling are post-processing methods (i.e., that can be applied in post-processing of forecast data), we are left with two options to explore. We can recalibrate the component forecasters and then ensemble (C-E), or ensemble the components and then recalibrate (E-C). In the C-E model, we train ensemble weights in a leave-one-season-out format, on the recalibrated component forecasts. In the E-C model, we train ensemble weights in a leave-one-season-out format on the original component forecasts, and then recalibrate the ensemble forecasts.

Fig 9 reveals that E-C model performs better than the C-E model. This is in line with established forecasting theory, which states that linear ensembles (which take a linear combination of component forecasters, such as the FluSight ensemble approach) themselves are generally miscalibrated, even when the individual component forecasters are themselves calibrated [5, 8, 22].

## 4 Discussion

Even in a domain as complex as epidemic forecasting, relatively simple recalibration methods such as those described in this paper can significantly improve both calibration and accuracy. A forecaster's performance for any proper score can be decomposed into three components: the inherent uncertainty of the target itself, the resolution of the forecaster (concentration of the forecasts), and the reliability of the forecaster to the target (calibration) [1]. In epidemic forecasting, without seasonality-aware recalibration training (such as that proposed and implemented in this paper), recalibration will not affect the resolution term, which is left to the

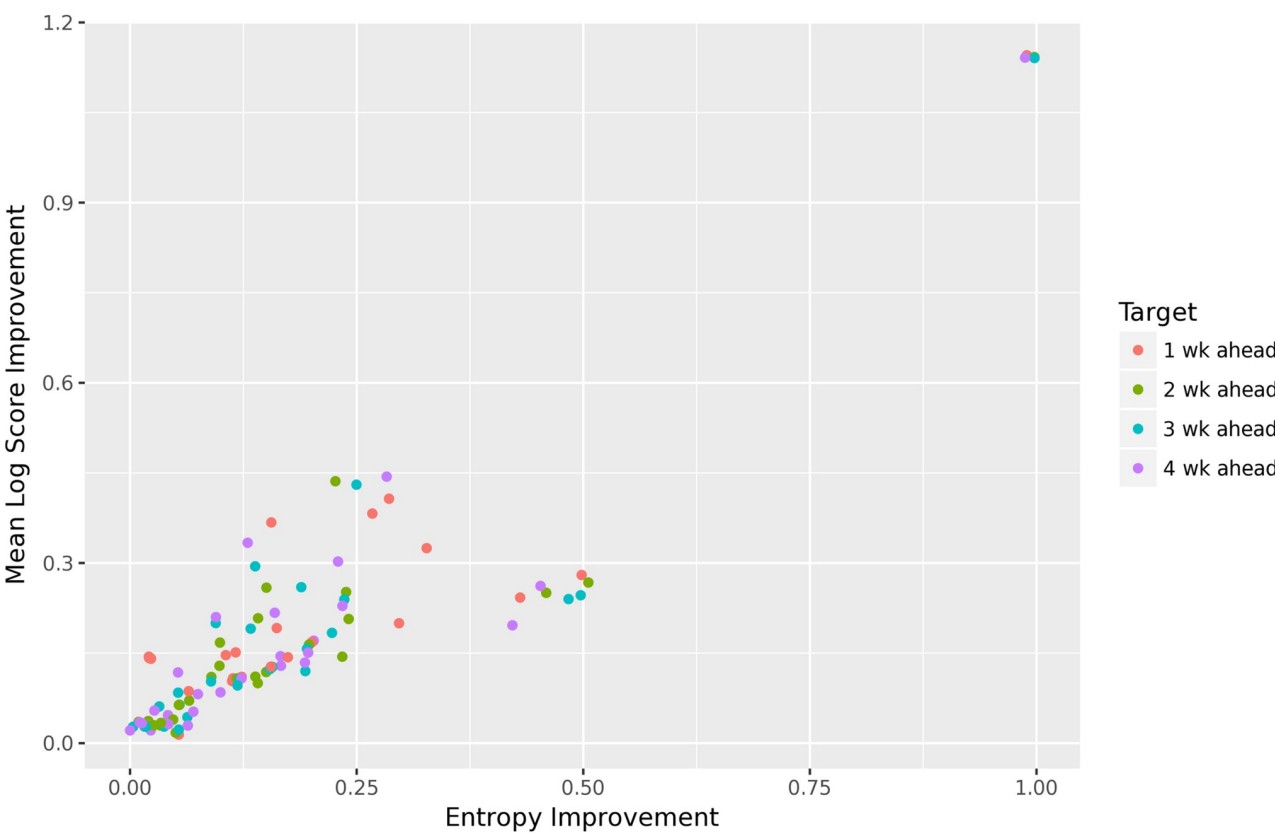

**Fig 6. Improvement in mean log score versus improvement in entropy for each of the 27 FluSight forecasters and short-term targets.** There is a clear linear trend (with slope approximately 1) between the improvement in calibration and the improvement in accuracy.

individual forecasters, but it will improve the reliability term. However, using seasonality-aware recalibration, it can also improve the resolution term.

Over 9 seasons of forecast data from 27 forecasters in the FluSight Challenge, we found that recalibration was especially helpful for the short-term targets (1–4 week ahead forecasts). With the exception of two very similar forecasters that have poor performance, the ensemble recalibration method was able to reduce the entropy of the PIT distribution to nearly zero (not or barely statistically significantly different than a uniform distribution). The recalibrated forecasts are therefore more accurate and more reliable. This is true across a diverse set of forecasters, including mechanistic, statistical, baseline, and ensemble models; indeed, as our recalibration method treats the forecaster as a black box, it can be applied to any forecaster, given access to suitable training data (retrospective historical forecasts).

Recalibrating influenza forecasts avoids challenges present in other forecasting environments, such as nonseasonality, a lack of consistent forecasters spanning many seasons, and little training datalittle training data, nonseasonality, and consistent forecasting models spanning many seasons. Although this makes recalibrating influenza forecasts a relatively easier task, we believe that this recalibration method can be applied to forecasting other diseases as well. For example, dengue fever is a seasonal disease with training data since 2014 available for forecasting [23]. Aedes mosquito counts are another seasonal target of interest to the CDC, which has released several years of training data for some counties for the purpose of forecasting [24]. This recalibration method, with its seasonal component, can be applied to these forecasts.

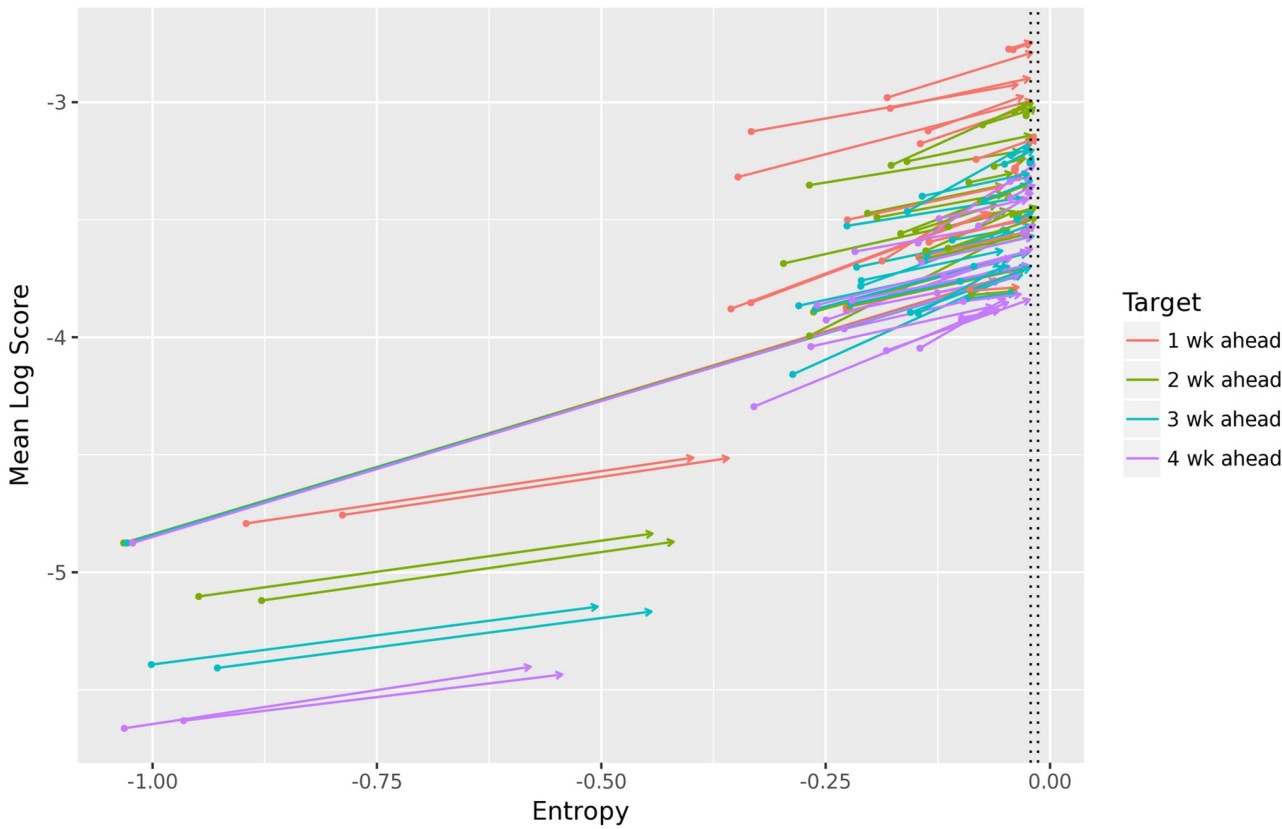

**Fig 7. Entropy and mean log score before and after recalibration, for each of the 27 FluSight forecasters and short-term targets.** The tail of arrow represents a quantity before recalibration, and the head after recalibration. The dotted lines show the central 90% interval of the entropy of a comparably-sized sample of standard uniform random variables for comparison. For all but two forecasters (the eight bottom-most line segments), the ensemble recalibration method achieves almost perfect calibration as evidenced by a near-zero PIT entropy, and this is accompanied by significant improvements in accuracy.

In application to nonseasonal diseases, such as COVID-19 (currently), this method can easily be modified to use all available PIT values, as opposed to the selective training used for influenza forecasts. Alternatively, selective training could be done not by calendar week but by some other feature(s) that differentiates a forecaster's behavior (e.g., whether cases are increasing or decreasing). While this allows for a flexible approach to recalibrate a variety of seasonal and nonseasonal diseases, this may be difficult to implement effectively in practice. In other cases where the PIT distribution changes slowly over time, training could be done only on the most recent forecasts to improve the estimate of $\hat{G}$. This selective training approach has been successful in recalibrating COVID-19 forecasts [25]. The ensemble approach allows for the incorporation of multiple models trained on different historical forecasts, or even different recalibration methods altogether.

Regarding a lack of consistent forecasters, even if a forecaster has been modified continuously over many years and previous performance is not indicative of current performance, recalibration can be trained on retrospective forecasts produced by the current forecaster.

A lack of training data is a more difficult problem to solve. An obvious problem of limited training data is the variance in estimating $\hat{G}$, but an additional challenge is that it is difficult to confirm our assumption that the PIT distribution is stationary over time. If we cannot detect

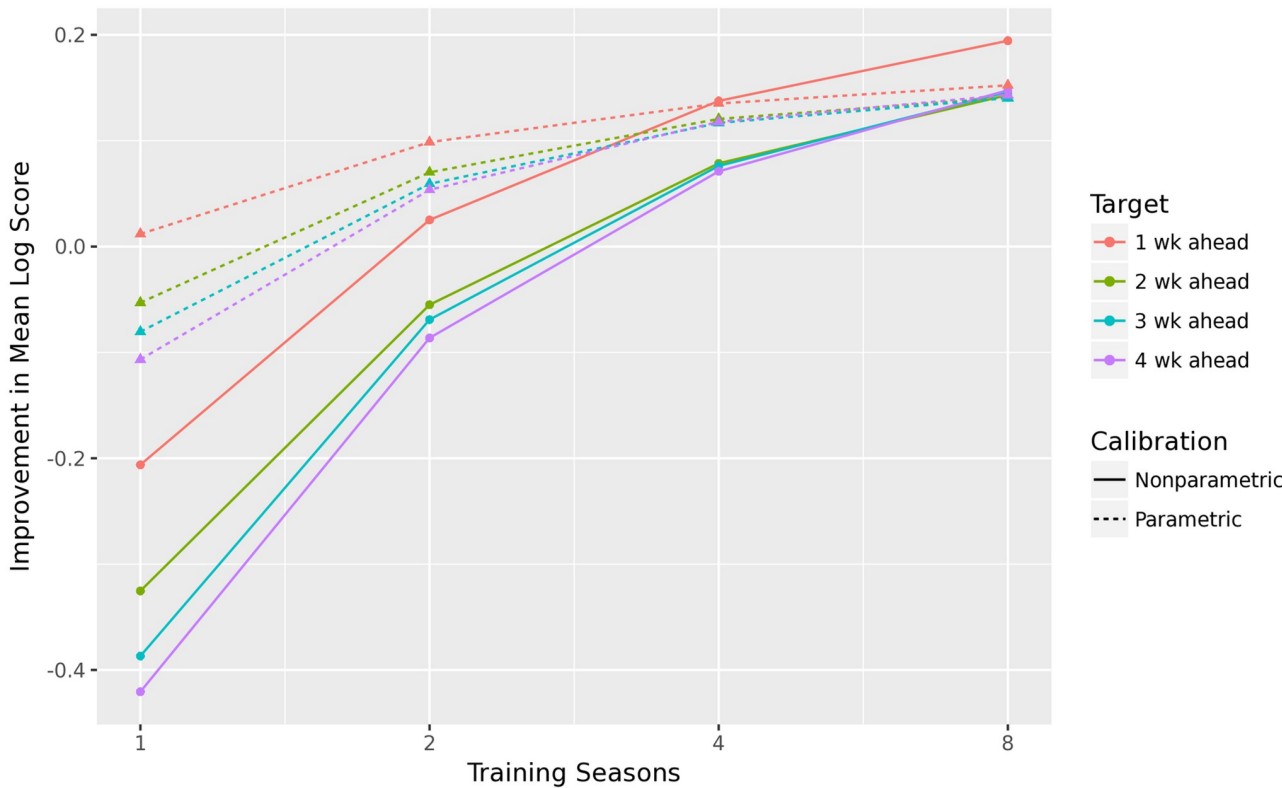

**Fig 8. Improvement in mean log score after recalibration, averaged over all 27 FluSight forecasters, by number of training seasons.** We perform three runs for each of the nine available seasons and $n \in \{1, 2, 4, 8\}$, where a run consists of randomly sampling $n$ other seasons to train recalibration for each of the 27 FluSight forecasters. Each point in the plot is averaged over $9 \times 3 = 27$ runs. As expected, the parametric method is more robust to limited training data than the nonparametric method.

that the PIT distribution changes over time, we will make inappropriate "corrections" to the forecasts that could harm calibration and accuracy. In practice, recalibration improved performance of the FluSight Challenge forecasts with relatively little training data, as shown in Fig 8. In less well-behaved applications, however, performance could decrease. We have made these recalibration methods available online so that a user can experiment with his or her own forecasts and determine whether or not recalibration improves performance [21].

The performance of recalibration with respect to the seasonal targets (onset, peak week, and peak percentage) was less conclusive than that of the short-term targets. Although the mean log score averaged over all of the forecasters was improved, recalibration only improved the performance of about three-quarters of the forecasters. Seasonal targets are inherently more difficult to recalibrate because at the end of the season, the true value has almost certainly been observed, and the forecasts are highly confident. For these forecasts, the correct bin has a mass of almost 1, and the observed PIT value then is approximately 0.5. At the end of the season, the PIT distribution is very concentrated at 0.5, which indicates underconfidence and poor calibration. If these PIT values of 0.5 are used to train forecasts for recalibration earlier in the season, before the target is observed, then recalibration incorrectly makes the forecast more confident. Because one is unsure whether the season peak has occurred or not for several weeks after the peak occurs, recalibration training is a nontrivial task. In general, more work is required to reliably improve accuracy and calibration for seasonal targets, which is a topic for future work.

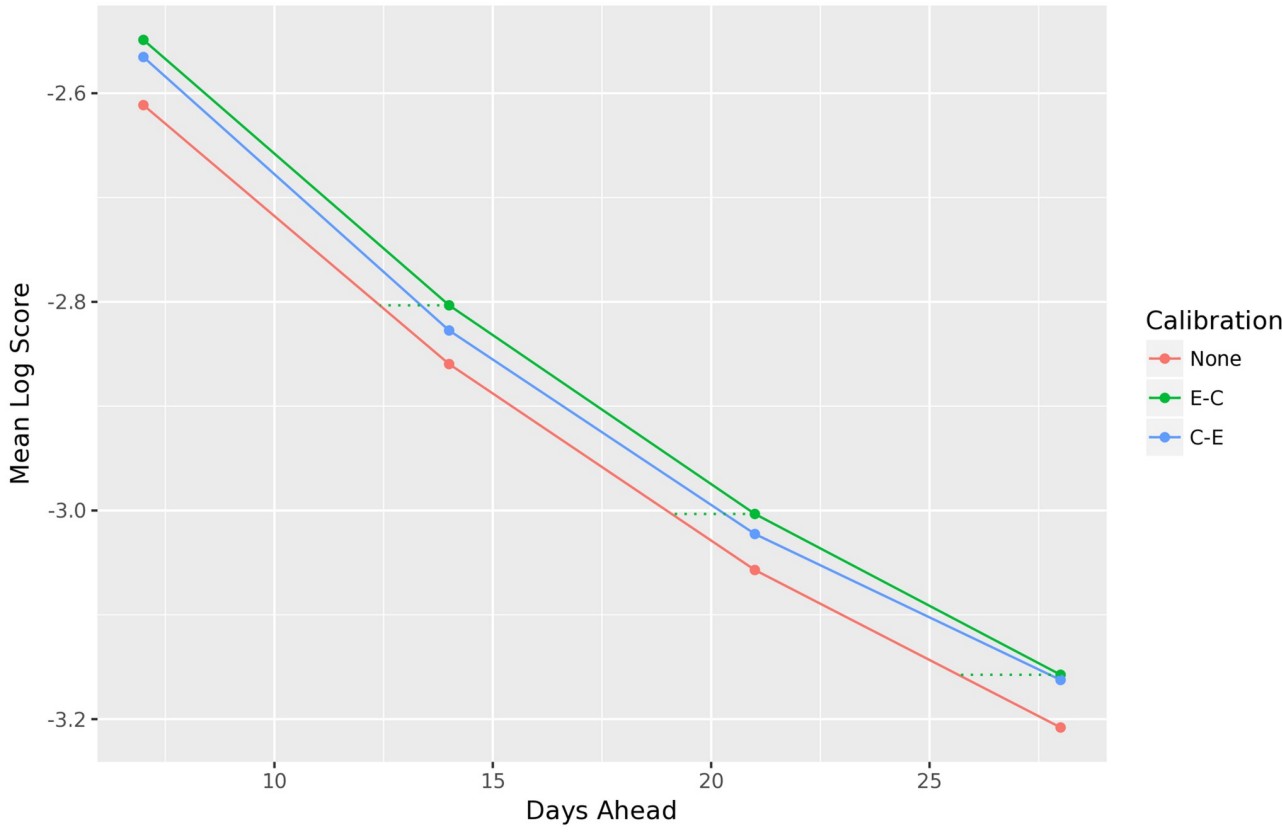

**Fig 9. Mean log score for the two different approaches to recalibrating the FluSight ensemble forecaster, with C-E and E-C reflecting the order of recalibration and ensembling.** Both the C-E and E-C models outperform the original ensemble (with no recalibration), but ensembling followed by recalibration performs best. By viewing forecast performance as a function of time, recalibration increases performance as much as roughly two days' time would.

## Supporting information

**S1 Appendix. The supporting information contains additional results.**
(PDF)

## Author Contributions

**Investigation:** Aaron Rumack.

**Methodology:** Aaron Rumack.

**Software:** Aaron Rumack.

**Supervision:** Ryan J. Tibshirani, Roni Rosenfeld.

**Writing – original draft:** Aaron Rumack.

**Writing – review & editing:** Aaron Rumack, Ryan J. Tibshirani, Roni Rosenfeld.

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
