## [Decision Letter · Decision Letter 0]

16 May 2022

Dear Mr. Rumack,

Thank you very much for submitting your manuscript "Recalibrating probabilistic forecasts of epidemics" for consideration at PLOS Computational Biology.

As with all papers reviewed by the journal, your manuscript was reviewed by members of the editorial board and by several independent reviewers. In light of the reviews (below this email), we would like to invite the resubmission of a significantly-revised version that takes into account the reviewers' comments. In particular, both reviewers have commented on how influenza forecasts may be a special case of infectious disease forecasts because of marked seasonality, multiple years worth of data, and lack of changes in the structure of individual models (so that adjusting from past performances can be particularly useful). Accordingly, they have provided suggestions for sensitivity analyses. We would also want this limitation to be highlighted in the discussion.

We cannot make any decision about publication until we have seen the revised manuscript and your response to the reviewers' comments. Your revised manuscript is also likely to be sent to reviewers for further evaluation.

Sincerely,

Cecile Viboud

Associate Editor

PLOS Computational Biology

Thomas Leitner

Deputy Editor

PLOS Computational Biology

Reviewer's Responses to Questions

**Comments to the Authors:**

Reviewer #1: Summary

The authors present a novel technique for recalibating black-box forecasters that uses the distribution of PIT values from past forecasts to improve measures of calibration as well as log scores of future forecasts. They demonstrate the effectiveness of their approach by showing the improvement achieved for seasonal forecasts of influenza. Their proposed approach in a first step estimates PIT distributions (using both a parametric and a non-parametric method) and uses it to recalibrate forecasts. In a second step, they create a weighted ensemble of these two approaches and an unmodified forecast. To my knowledge this approach is novel and is one that I find genuinely exciting. I think the manuscript could be made even better by addressing a few points.

Major comments

First, the FluSight data set chosen to me feels like an idealised example, but this is not discussed in the paper. The authors use data that would not be available to researchers in many real-time settings, for example by using all other seasons, even those that lie in the future, to recalibrate forecasts for a given season. The data set is also very rich, including forecasts from the exact same models available across 9 seasons. In many settings, however, models evolve over time and it may not be possible to create retrospective forecasts. Diseases also may be non-seasonal or a disease may be seen for the first time in a given region (e.g. Ebola, Chikungunya in the Americas, COVID-19). I think it would be important to make these particular features of the example data clear and discuss in greater detail in which other settings the recalibration approach can be applied and where it may fail.

I suggest showing how well the recalibration method worked if only data from a single season were available. The data collated by the COVID-19 Forecast Hub could also make for an interesting example.

Secondly, I think Figure 3 and Figure should be made more consistent and captions should to be more detailed, as it is sometimes not clear what the figure shows. In particular:

- Figure 3: I'm not sure I exactly understand what was done. Why was the log score of the most accurate forecaster chosen, rather than say the average of all forecasters? What forecast horizon was used? Maybe it would be helpful to clearly separate k and 'window size' as at least to me they were confusing at first. Do I understand correctly, that k = 10 means 10 weeks prior + 10 weeks past the week, so the window is actually 20 weeks? when k = 10, how do you deal with the forecasts in the first 10 weeks (or the last ones). Is the window shortened for these, or are they omitted from scoring? In addition: maybe it would be interesting to extend the shown window size even further, until we actually see performance degrade.

- Figure 4: It is slightly confusing that the non-parametric method performs about equally well for 2-, 3-, and 4-week-ahead forecasts in this plot. Just looking at Figure 3, one would expect a negative effect for 4-week-ahead forecasts and a 3-week window size. I assume this is because Figure 4 uses all 27 forecasters and Figure 3 only the best one? To me this is confusing and it would be helpful if the Figures were more consistent.

Thirdly, the code, as of now, is not very accessible. It currently sits on a branch of a much greater repository and it is unclear a) whether this code is meant to stay there indefinitely, b) how the code ties in with the greater context of the repository (what is required for the recalibration and what is not) and c) how someone could use the code for their own work. I think it would greatly enhance accessibility of the code if the authors could a) move it to a separate repository (or maybe a subfolder of the main repository), rather than just a branch of an existing repository, b) document the code and its functions in greater detail and c) provide some more explanations on how to run the code.

Minor points

- Introduction: Maybe it would be a good idea to mention the forecasting paradigm to maximise sharpness subject to calibration (see Gneiting et al., Probabilistic forecasts, calibration and sharpness, 2007)?

- Line 10: If calibration is one aspect, which are the other two? This is explained in the discussion, but maybe could also be included here.

- Line 58: I asked myself what the the added complexity was if CDFs were not strictly increasing. Maybe either remove "for simplicity" or add a short explanation of why this is necessary?

- Line 59: I'm not sure why the Brier Score which is usually associated with binary forecasts is mentioned here. Maybe this could be made a little more clear (or removed).

- A few lines below line 60 (there are no line numbers here unfortunately): The observation was previously called $x_i$, (and $y_i$ was the random variable). It is maybe slightly confusing that the observation is now called $y_i$, where before the observation was $x_i$.

- Line 70: Maybe mention that a uniform PIT is a necessary, but not a sufficient condition for probabilistic calibration. Gneiting et al. (Probabilistic forecasts, calibration and sharpness, 2007) and Hamill (Interpretation of Rank Histograms for Verifying Ensemble Forecasts, 2001) mention examples of forecasters who are mis-calibrated, but have uniform PIT histograms.

- Equation 2: Again $x_i$ and $y_i$ are used differently than before

- Line 144 "too conservative" is slightly ambiguous. What it means is that forecasters don't adapt to trends quickly enough, but it could also easily be understood as "forecasts are to wide and uncertain"

- I unfortunately did not really understand the sentence in lines 178-180. Maybe the exact format could be worded more clearly?

- line 185: In applied setting, would't it make more sense to use leave-future-out CV? Currently this is using data that would not have been available to researchers making forecast at the time. This ties into the point made in the summary above: I would really like to know what performance is in less idealised circumstances.

- I'm not entirely sure I understood the full algorithm explained from line 187 on. What is confusing to me is that seasons are denoted with r and s, and weeks with $i$, but then later on (e.g. line 193), seasons are also denoted with $i$. I am also not sure I understood the difference between step 1 and step 3.

- Maybe it would be helpful to point out whether the weeks $i$ in the seasons calendar correspond to calendar weeks, or are they relative to something?

- 218 - 19: This sentence implies that the amount of improvement is close between the two, but rather I understand the proportion of improved models is close.

Reviewer #2: Major Issues

The paper gives a nice discussion (Sec 2) of the too-often-overlooked recalibration issue for influenza forecasting.

Relative to the 4-step recipe for the recalibration approach in Section 3, there appear to be implicit assumptions that should hold for the approach to be reasonable. For example, Step 1 defines a set of relevant (essentially exchangeable?) forecast errors for a "current" forecast, and it relies strongly on a seasonal structure. Despite certain claims (as in Sec. 2.6 and "the seasonal nature of epidemic forecasting"), not all highly communicable diseases are intrinsically seasonal, let alone do they have ample seasonal data available for recalibration. Examples: SARS, ebola, many animal diseases, and to this point in time, covid. The seasonal assumption clearly applies to influenza, but may not generalize easily to other forecasting environments.

Next, it appears implicit that a forecaster's prediction errors are fundamentally static. Stated less charitably relative to your FluSight example, were all forecasters so oblivious to their forecasting performance over the multi-year forecasting time frame that they never learned anything that helped them improve their modeling? Do the results of your analysis have no value to them beyond modifying prediction intervals after the fact? At a minimum, shouldn't this static assumption (i.e., that forecasters can never learn nor modify their behavior) be somehow confirmed before the recalibration approach is routinely applied? As in the previous paragraph, the assumption appears to give good results for influenza, but may(?) not generalize to other dynamic forecasting environments.

A suggestion is that underlying assumptions such as exchangeability and static forecasts be made explicit. Further, if the authors truly feel that their approach works as well for other infectious diseases as it does for influenza, some justification of this claim together with concrete examples would be a welcome addition to the paper. Otherwise, the emphasis of the paper (as well as its title) should be modified to reflect the influenza-centric nature of the approach.

Minor Issues

I was struck by the comment on p.1 in the summary that "epidemic forecasting is a relatively new field". Different points of view are not hard to find, e.g., "failure in epidemic forecasting is an old problem" (Ioannidis, Cripps, and Tanner; International Journal of Forecasting, 2021, Sec. 1). Indeed, the long/undistinguished history of epidemic forecasting is actually good motivation for your work. Or maybe there's some confusion over what your use of the word "new" means?

I was disappointed at the lack of an overall discussion of the optimized weight values { w_i } for each ILI forecaster/target. What do the optimized weights imply about the three components of the ensemble (Sec 2.5) and/or about the individual forecasters? Could some detail be added on this subject?

I liked the comment "recalibration only improved the performance of about three-quarters of the forecasters" on seasonal targets as it provided some insight to the analysis. Unfortunately, the discussion of this issue in the last paragraph of Sec. 4 did little to distinguish which types of forecasters gain by using the approach and which types don't. Was it that one-quarter of FluSight forecasters were well calibrated to begin with, so that no major improvement was possible? Did the aforementioned one-quarter of forecasters modify their predictive model over time, invalidating the implicit assumptions? Or ...? Providing more intuition as to when the approach works well (and when it doesn't) would be a nice addition to the paper.

Most influenza forecasters have a reputation of being overconfident, with U-shaped PIT curves. Does residual overconfidence still exist for your recalibrated forecasts, though to a much lesser degree? Is the approach mainly a bias correction activity?

Lastly, could the paper could note that the approach can be easily adapted to incorporate other recalibration methods beyond just those considered in the paper.

**Have the authors made all data and (if applicable) computational code underlying the findings in their manuscript fully available?**

Reviewer #1: Yes

Reviewer #2: None

PLOS authors have the option to publish the peer review history of their article (what does this mean?). If published, this will include your full peer review and any attached files.

Reviewer #1: **Yes: **Nikos I. Bosse

Reviewer #2: No
---

## [Decision Letter · Decision Letter 1]

12 Aug 2022

Dear Mr. Rumack,

Thank you very much for submitting your revised manuscript "Recalibrating probabilistic forecasts of epidemics" for consideration at PLOS Computational Biology. The revised manuscript went back to the initial reviewers. Both reviewers agree that the new version is substantially improved and clearer. Yet, the second reviewer has asked that you cast a more critical eye on the stationarity of PIT and be more specific about the type of non-seasonal diseases raised in the discussion. Based on the reviews, we will accept this manuscript for publication, but we would very much appreciate if you could address/caveat these points in the final version of the manuscript.

Sincerely,

Cecile Viboud

Associate Editor

PLOS Computational Biology

Thomas Leitner

Deputy Editor

PLOS Computational Biology

[LINK]

Reviewer's Responses to Questions

**Comments to the Authors:**

Reviewer #1: Thank you very much for addressing our comments. I would be very happy to see this work published.

Reviewer #2: In my previous review, I asked you to give specific, concrete examples of seasonal diseases (besides influenza) for which currently available public health data allow for the use of your methodology. I was disappointed that you couldn’t/wouldn’t provide a single such example. The paper would be much improved if its sweeping generalizations about applicability (such as lines 296-297) were accompanied by some detailed supporting evidence.

In any case, the approach has potential benefit for influenza forecasting. That some of these predictive models have been used for as long as they have and still aren’t well calibrated is somewhat embarrassing to the epi community -- especially in light of recent high-profile problems with poor/no uncertainty estimates for COVID-19 forecasts (see the reference cited in my previous review) and the history of other epidemiological probabilistic assessments, such as questionable confidence intervals for mechanistic model parameters (see, e.g., general work by Ioannidis, Tong, and others). Recalibration of probabilistic statements is topical and important.

Also on a positive note, the basis for the approach is more clearly presented in the revision. Adding discussion in Section 3.3 on varying amounts of training data is a nice touch, although the text is misleadingly one-sided. The good news, as is well summarized, is that there is only modest loss of precision with limited training data so long as the modeling assumptions are valid. The bad news, however, needs attention: having limited training data means having limited ability to meaningfully check the validity of those assumptions in the first place, which undermines “robustness” (line 247) and could easily lead to poor results from the approach.

Specifically, the stationarity assumption (lines 148-149) that the PIT distribution is the same for the training and test sets should not be taken lightly, especially for nonseasonal diseases. Consider this assumption relative to the most prominent nonseasonal disease at the moment, COVID-19, where the PIT distribution evolved over time for a variety of reasons. This type of nonstationarity can be circumvented by using only the most current data for recalibration, similar to the rationale in your lines 156-164 (aside: such an approach has recently been detailed -- Picard and Osthus, medRxiv 2022; even then, daily data might be required for true success in practice). Although the COVID-19 paradigm is nearly antithetic to the static-seasonal-disease, static-forecast-model paradigm you focus on, it emphasizes the importance of having adequate training data to check modeling assumptions.

The added paragraph (lines 296-311) on nonseasonal diseases is very weak. Readers will immediately wonder: exactly which nonseasonal diseases are you talking about? What disease-specific data sets exist for forecasting, now or in the foreseeable future? What’s so special about an “entire year” (line 303) for a disease that doesn’t care about the calendar? Because these questions aren’t addressed, readers will find this paragraph to be exasperatingly devoid of any substantive details. There’s a case to be made for your assertions, but you’re not making it. A related point: the paper’s approach is somewhat ahead of its time, awaiting a world where timely, standardized, multi-year, and easily accessible training data are available for all seasonal and nonseasonal diseases of interest. Admittedly, public health and biosurveillance data are moving (very slowly) in this direction, but they’re nowhere near that state now.

**Have the authors made all data and (if applicable) computational code underlying the findings in their manuscript fully available?**

Reviewer #1: Yes

Reviewer #2: None

PLOS authors have the option to publish the peer review history of their article (what does this mean?). If published, this will include your full peer review and any attached files.

Reviewer #1: **Yes: **Nikos I. Bosse

Reviewer #2: No

Figure Files:

Data Requirements:

Reproducibility:

References:

---

## [Decision Letter · Decision Letter 2]

28 Nov 2022

Dear Mr. Rumack,

We are pleased to inform you that your manuscript 'Recalibrating probabilistic forecasts of epidemics' has been provisionally accepted for publication in PLOS Computational Biology.

Best regards,

Cecile Viboud

Academic Editor

PLOS Computational Biology

Thomas Leitner

Section Editor

PLOS Computational Biology

Feilim Mac Gabhann

Editor-in-Chief

PLOS Computational Biology

Reviewer's Responses to Questions

**Comments to the Authors:**

Reviewer #2: none

**Have the authors made all data and (if applicable) computational code underlying the findings in their manuscript fully available?**

Reviewer #2: None

PLOS authors have the option to publish the peer review history of their article (what does this mean?). If published, this will include your full peer review and any attached files.

Reviewer #2: No

---

## [Editor Report · Acceptance letter]

12 Dec 2022

PCOMPBIOL-D-21-02239R2 

Recalibrating probabilistic forecasts of epidemics

Dear Dr Rumack,

I am pleased to inform you that your manuscript has been formally accepted for publication in PLOS Computational Biology. Your manuscript is now with our production department and you will be notified of the publication date in due course.

With kind regards,

Zsofia Freund
